# Association of cataract and sun exposure in geographically diverse populations of India: The CASE study. First Report of the ICMR-EYE SEE Study Group

Praveen Vashist[1]◉, Radhika Tandon○[1]◉*, G. V. S. Murthy[2]◉, C. K. Barua[3], Dipali Deka[3], Sachchidanand Singh○[4]◉, Vivek Gupta○[1], Noopur Gupta[1], Meenakshi Wadhwani[1], Rashmi Singh[1], K. Vishwanath[5], on behalf of the ICMR-EYE SEE Study Group[¶]

**1** Community Ophthalmology, Dr RP Centre for Ophthalmic Sciences, New Delhi, India, **2** Public Health Foundation of India, Hyderabad, Telangana, India, **3** Regional Institute of Ophthalmology, Guwahati, Assam, India, **4** National Physical Laboratory, New Delhi, India, **5** Pushpagiri Vitreo Retina Institute, Secunderabad, Telangana, India

◉ These authors contributed equally to this work.
¶ Membership of the ICMR-EYE SEE Study Group is provided in the Acknowledgements.
* radhika_tan@yahoo.com

**Data Availability Statement:** Data are within the Supporting Information files and available from the DRYAD repository: Vashist, Praveen et al. (2019),

## Abstract

### Purpose

To determine the prevalence of cataract and its association with sun exposure and other environmental risk factors in three different geographically diverse populations of India.

### Design

Population based cross sectional study during 2010–2016

### Participants

People aged ≥ 40 years residing in randomly sampled villages were enumerated (12021) and 9735 (81%) underwent ophthalmic evaluation from plains, hilly and coastal regions (3595, 3231, 2909 respectively)

### Methods

A detailed questionnaire-based interview about outdoor activity in present, past and remote past, usage of sun protective measures, exposure to smoke, and detailed ophthalmic examination including assessment of uncorrected and best corrected visual acuity, measurement of intraocular pressure, slit lamp examination, lens opacities categorization using LOCS III and posterior segment evaluation was done. Lifetime effective sun exposure was calculated using Melbourne formula and expressed as quintiles. These were supplemented with physical environmental measurements.

Association of cataract and sun exposure in geographically diverse populations of India: The CASE study. First Report of the ICMR-EYE SEE Study Group, v2, Dryad, Dataset, https://doi.org/10.5061/dryad.5qfttdz19

**Funding:** Indian Council of Medical Research (ICMR) provided funding (Grant No 68/4/2009-NCD-1) for this investigator-initiated project and had no direct role in the design and conduct of study. PV, RT, GVSM, DD, SS are Principal Investigators of respective sites.

**Competing interests:** The authors have declared that no competing interests exist.

## Main outcome measures

Lifetime sun exposure hours, smoking, indoor kitchen smoke exposure and their association with cataract and subtypes. Prevalence of cataract calculated based on lens opacities or evidence of cataract surgery.

## Results

Cataract was identified in 3231 (33.3%) participants. Prevalence of cataract in males (32.3%) and females (34.1%) was similar. Nuclear cataract was the commonest sub-type identified in 94.7% of affected eyes. Sun exposure had a significant association with cataract with odds ratio (OR) increasing from 1.6 (95% Confidence Intervals [CI]: 1.4, 1.9) in 3rd quintile, to 2.6 (CI: 2.2, 3.1) in 4th quintile and 9.4 (CI: 7.9, 11.2) in 5th quintile (p<0.0001). Cataract also showed a significant association with smoking (OR: 1.4, CI: 1.2, 1.6) and indoor kitchen smoke exposure (OR: 1.2, CI: 1.0–1.4). Nuclear cataract showed a positive association with increasing sun exposure in 3rd (β coefficient 0.5, CI:0.2–0.7), 4th (β: 0.9, CI: 0.7–1.1) and 5th (β: 2.1, CI:1.8–2.4) quintiles of sun exposure, smoking (β: 0.4, CI: 0.2–0.6) and indoor kitchen smoke exposure (β: 0.3, CI: 01–0.5) while cortical cataract showed a positive association with sun exposure only in 5th quintile (β: 2.6, CI:1.0–4.2). Posterior sub-capsular cataract was not associated with any of the risk factors.

## Conclusion

Cataract is associated with increasing level of sun exposure, smoking and exposure to indoor kitchen smoke.

## Introduction

Cataract remains the most important cause of blindness globally and in India. In India the onset of cataract is reported to occur a decade earlier as compared to the Western population. [1] Efforts for elimination of blindness due to cataract largely focus on surgical management. Prevention of cataract requires an understanding of the epidemiological risk factors and the recognition of modifiable risk factors among them. Internationally, there is evidence to support the role of sun exposure in development of cortical cataract. [2,3] Though various studies have tried to address the role of potential environmental and behavioural risk factors such as smoking and household smoke exposure for development of cataract, the variable findings drive a need for further exploration in this area.[4] There is limited evidence on the association of risk factors pertaining to the sub types of cataract, especially in low and middle income countries. Outdoor activity or sun exposure was identified as a risk factor for cortical cataract (CC) in China, cortical and posterior sub capsular cataract (PSC) in USA and nuclear cataract (NC) in Australia. [2,3,5] The purpose of this study was to determine the association of sun exposure and other risk factors with various types of cataract in populations from three geographically distinct regions of India.

## Materials and methods

This study was conducted in compliance with the guidelines in the declaration of Helsinki. The study was approved by Institutional Ethics Committee of All India Institute of Medical

Sciences, New Delhi (P-16/04.08.2009); Indian Institute of Public Health Hyderabad (33/2011–08–08); and Regional Institute of Ophthalmology, Guwahati (MC/190/2007/1098-23.02.2010). Written informed consent was taken from each participant.

## Study design and location

This was a population based cross sectional study conducted in in three different rural areas of India: plain area in north, hilly area in north-east, and coastal area in south during 2010 to 2016. Villages in Gurgaon district located in National Capital Region of Delhi represent the northern plains; Prakasam district in Andhra Pradesh, the southern coastal, and; Guwahati, the eastern hilly regions. NCR Delhi has an altitude of 216 metres above sea level and has a monsoon-influenced composite climate. Prakasam is primarily a coastal area located adjacent to the Bay of Bengal with a mean elevation of 10 metres and bears a tropical climate. Guwahati is located in the eastern Himalayan belt and the villages are settled in diverse topographical areas comprising of plains, riverine or char areas as well as settlements along the hill slopes at higher altitude (range 50–680 metres).

## Study population and sampling

Residents of the study area, aged 40 years or above were eligible to participate. The study aimed at enrolling 3500 participants aged $\geq$ 40 years from each location and response rate of around 85% selected using cluster sampling. Using census village level population data, we divided villages into smaller clusters of 400–600 population each having 100–150 eligible participants and by simple random sampling, 35 clusters were identified. The survey team visited the village prepared a map, and identified segments of 400–600 population. One cluster segment was then selected by draw of chits. All the residents aged 40+ in this segment were invited to participate in the study. In case of refusals or non-availability, participants were re-contacted up to three times.

House visits were conducted by trained field workers and participants were interviewed using a structured questionnaire. It included questions on socio-demographic information, smoking, indoor kitchen smoke exposure, and sun exposure. All participants above 40 years of age were asked to come for detailed ophthalmic examination at a locally arranged clinic.

## Assessment of risk factors

**Sun exposure.** Information was obtained on hours spent outdoors in different periods of life, and for each period, duration (years) of that period, average hours per day spent outdoors during the day (8a.m. to 5p.m.), use of protective head gears (umbrella, dupatta, hat, sunglasses, cap, others), and hours for which head gear was used. The effective mean lifetime sun exposure was estimated using a modified Melbourne study formula as below:

## Lifetime effective sun exposure

$$\text{OEeff} = \sum_{i=1}^{p} \left( (H_i \times 365 \times Y_i) + \left( \sum_{j=1}^{n} H_{ij} \times F_j \times 365 \times Y_i \right) \right)$$

where, *OEeff* = lifetime effective sun exposure; $i$ = period of life; $Y_i$ = duration of period '$i$' in years; $H_i$ = hours of sun exposure without head gear usage in period '$i$'; $j$ = head gear used in period '$i$'; $H_{ij}$ = hours of sun exposure with head gear '$j$' in period '$i$'; $Fj$ = sun protection factor for headgear '$j$'.

The sun-protection factors of ocular protection applied in the calculation are 0.53 for facial cloth protection (veil, towel, etc), traditional headgear (pagdi, mundas, towels), umbrella, and caps/hats; and 0.21 for sunglasses.[6]

**Ultraviolet, aerosol exposure.** The measurements of total (direct + diffuse) UVA (315–400 nm), UVB (280–315 nm) flux were done at Delhi during October 2012 to September 2015 and compared with the satellite-based Clouds and Earth's Radiant Energy System (CERES) data products for UVA, UVB to validate the same. The daily mean UVA and UVB measurements showed excellent agreement (r ~0.92–0.93) with satellite-retrieved CERES UV fluxes. The mean bias errors for UVA and UVB with respect to NPL-measured UV fluxes are about -1% and 9%, respectively. More details about the comparison of CERES data with observation can be found elsewhere [7] Similarly, the aerosol optical depth (AOD) data measured from Delhi have been thoroughly compared with the satellite derived AOD values[8]As the UV and AOD data estimated from the satellite products were already verified from the ground based measurements, it has been used for the long term UVA, UVB and AOD values in the present study (2010 to 2016) at the three locations, Delhi, Guwahati and Vishakhapatnam representing the three regions of India described earlier, Vishakhapatnam being a close station to Prakasam.

## Ocular examination

This was conducted by trained ophthalmologists and optometrists. The kappa for inter-observer variation was more than 0.8. This included visual acuity examination using Tumbling E of the Early Treatment of Diabetic retinopathy Study (ETDRS) chart and recorded as Snellen equivalent ($\geq$ 4 of 5 letters correctly identified in each row) separately for each eye. If the visual acuity in either of the eye of the participant was worse than logarithm of minimal angle of resolution 0.3, an optometrist conducted refraction manually and with auto-refractometer (PRK-5000, POTEK-KOREA) and the best corrected visual acuity was recorded. Pupillary dilatation to $\geq$ 6mm was achieved using 1% tropicamide after anterior segment biomicroscopy. A clinical examination of each eye was performed, which included anterior and posterior segment examination through distant direct ophthalmoscope, direct and indirect ophthalmoscopy and biomicroscopic examination using slit lamp biomicroscopy and intraocular pressure was measured using portable non-contact tonometer (Reichert PT 100).

## Grading of lens opacities

Lens was examined for presence of cataract and grade of cataract using LOCS III figures after dilatation of pupil.[9] Patients with traumatic cataract, developmental cataract, aphakia and pseudophakia were excluded from LOCS examination. The standard set of photographs were mounted next to the slit lamp for grading. The nuclear cataract was graded for both nuclear colour and opalescence, from 0.1–6.9 for NO/NC, cortical cataract was graded on a decimal scale of 0.1–5.9 according to the opacity that obscured the light reflex on retroillumination. Posterior subcapsular cataract was graded on a decimal scale of 0.1–5.9 only if the opacity in posterior capsule is visible against red reflex. In absence of opacity, a score of zero was given. Cataract was graded based on LOCS III grade in the worse eye of $\geq$2 for nuclear cataract (either of nuclear opalescence or colour score), cortical and posterior subcapsular cataract.

## Definitions

Person with cataract: Presence of lenticular opacities on clinical examination, aphakia or pseudophakia in any eye while excluding traumatic, developmental cataracts.

Presence of cortical cataract: LOCS III Cortical Opacity Score ≥ 2.0. Similar definitions were used for nuclear cataract and PSC.

## Statistical analysis

Double entry of all data was done in a Microsoft Access™ database and they were matched to identify and correct transcription errors. Data were analyzed using Stata 13 (StataCorp, College Station, TX). Participants with incomplete information on sun exposure or ocular examination were excluded. All study participants were distributed across quintiles, based lifetime effective sun exposure. The prevalence of cataract was assessed among persons whose clinical lens evaluation was complete. Prevalence of specific subtype of cataract was calculated among participants whose LOCS III evaluation in either eye was done. Participants with traumatic or developmental cataracts were excluded. The combination patterns of subtypes of cataracts in the LOCS III assessed eyes were tabulated. In case one eye had cataract while other eye was pseudophakic or apahakic, we classified the participant based on the cataractous eye. Multivariable logistic regression analysis was done to estimate the association of sun exposure, smoking, and gender and other risk factors with any cataract Analysis were repeated for each site and for pooled data. Patients with pure cortical, nuclear and PSC based on LOCS III score were identified and the association with sun exposure and other risk factors was evaluated using multinomial logistic regression analysis, keeping person with both eyes normal as controls. P value < 0.05 was considered statistically significant and 95% confidence intervals (CI) were calculated.

## Results

A total population of 12021 (4353, 4140, and 3528 in plain, hilly and coastal region respectively) was enumerated and 9735 (80.9%) participants completed risk factor evaluation and clinical examinations (80.9% overall, 82.6%, 78.01%, and 82.45% in plain, hilly and coastal region respectively). The mean age of participants was 54.5 years (SE 0.12), and 4426 (45.5%) were males. More than half of respondents (5000, 51.4%) were illiterate, highest being (66.2%) in the coastal area (Table 1). Among the respondents, 36.7% were not using any protective headgear against sunlight when outdoors during 9AM to 5PM. Most of the study population was using either veil/dupatta or saree or some form of traditional headgear (32.0%). Umbrella (2.46%), caps (1.54%) and sunglasses (1.06%) were used rarely.

### Prevalence of cataract

Of the examined persons, 6.1% were pseudophakic or aphakic in both eyes, and 6.5% in one eye. The overall prevalence of cataract was 33.3%. The prevalence of cataract was highest in Prakasam (42.4%) and lowest in Guwahati (26.6%). (**Table 2**) The prevalence increased with age, reaching a high of 90.1% in the 70+ years age group. The prevalence of cataract was similar across males (32.3%) and females (34.1%). Higher prevalence of cataract was observed among persons in 4th quintile (36.9%) and 5th quintile (66.4%) of lifetime effective sun exposure compared to lower exposures.

### Pattern of subtypes of cataract

Overall, the total number of eyes that had graded for cataract using LOCS III classification was 3475. Out of these, the nuclear cataract was the most common (94.7%) followed by cortical cataract (28.2%) and PSC (16.7%) (**Table 3**).

**Table 1. Demographic characteristics of study participants examined for the study.**

| | Delhi/Plain n (%) | Guwahati/Hills n (%) | Prakasam/Coastal n (%) | All Centres n (%) |
|---|---|---|---|---|
| | n = 3595 | n = 3231 | n = 2909 | n = 9735 |
| **Age** | | | | |
| Mean age (±SE) | 55.35 (0.20) | 53.39 (0.20) | 54.57 (0.21) | 54.46 (0.12) |
| Median (Min-Max) | 53 (40–99) | 50 (40–99) | 52 (40–99) | 52 (40–99) |
| **Gender** | | | | |
| Male | 1614 (44.90) | 1491 (46.15) | 1321 (45.41) | 4426 (45.46) |
| Female | 1981 (55.10) | 1740 (53.85) | 1588 (54.59) | 5309 (54.54) |
| **Education*** | | | | |
| Illiterate | 1769 (49.21) | 1306 (40.53) | 1925 (66.17) | 5000 (51.41) |
| Studied up to primary | 532 (14.8) | 779 (24.18) | 487 (16.74) | 1798 (18.49) |
| Middle School (6–8) | 471 (13.1) | 294 (9.12) | 169 (5.81) | 934 (9.6) |
| High School (9–12) | 721 (20.06) | 742 (23.03) | 262 (9.01) | 1725 (17.74) |
| Graduation | 102 (2.84) | 101 (3.13) | 65 (2.23) | 268 (2.76) |
| **Occupation*** | | | | |
| House work | 1712 (47.6) | 1528 (47.3) | 471 (16.2) | 3711 (38.1) |
| Unskilled | 801 (22.3) | 915 (28.3) | 1676 (57.6) | 3392 (34.8) |
| Skilled | 399 (11.1) | 396 (12.3) | 320 (11.0) | 1115 (11.5) |
| Unemployed | 683 (19.0) | 386 (11.9) | 439 (15.1) | 1508 (15.5) |
| **Lifetime cumulative effective sun exposure (hours)** | | | | |
| Median | 114140 | 72759 | 109889 | 96062 |
| Range (min.-max.) | 7305–314104 | 7305–223763 | 7305–252183 | 7305–314104 |

* Education status was not known for 9 participants in Guwahati and 1 in Prakasam. Occupation status was not known for 6 in Guwahati and 3 in Prakasam.

## Association of cataract with risk factors

A multiple logistic regression comparing the association of cataract with various risk factors, region wise and combined is shown in **Table 4**. In Delhi, the association of cataract was found to be stronger with the 4th (OR: 2.2, CI:1.6–3.0) and 5th (OR: 7.5, CI:5.5–10.4) quintile of sun exposure with the maximum association found with the 5th quintile of sun exposure. Similar results were seen in all the three regions. The people exposed to smoke in indoor kitchens had higher association with cataract (OR: 1.3, CI: 1.0–1.8). Smokers were found to have a higher association with cataract (OR: 1.4; CI: 1.1–1.6). Guwahati showed a positive association between smoking and cataract (OR: 1.5, CI: 1.1–1.9) but no association was seen with smoke exposure due to indoor kitchen (OR: 1.3, CI: 0.9–1.8). There was a positive association between cataract and 2nd to 5th quintiles of sun exposure with maximum being in the 4th (OR: 10.6, CI: 7.6–14.6) and 5th quintile (OR: 25.7, CI: 15.2–43.5) of exposure. Similarly, Prakasam showed a positive association of cataract with smoking (OR: 1.4, CI: 1.1–1.8), and with the 5th quintile of sun exposure (OR:5.0, CI:3.8–6.5). Guwahati and Prakasam were found to have more chances of development of cataract with OR: 2.0, CI: 1.7–2.3 and OR: 2.0, CI:1.8–2.3 respectively.

## Association of types of cataract with various risk factors

Participants having pure cortical, nuclear and posterior subcapsular cataracts were identified. Among them, association of cataract with various risk factors were evaluated using multinomial logistic regression (Table 5). Gender did not show any association with cataract. Nuclear cataract showed a positive association with smoking (β coefficient: 0.4; CI: 0.2–0.6)

**Table 2. Prevalence of cataract in the presence of various risk factors.**

| | Cataract Present | Normal Lens | Total |
|---|---|---|---|
| | n% | n% | n |
| **Total** | **3,228 (33.2)** | **6,483 (66.8)** | **9,711** |
| **Site** | | | |
| Delhi/Plain | 1,144 (31.9) | 2,443 (68.1) | 3,587 |
| Guwahati/Hills | 856 (26.6) | 2,368 (73.4) | 3,224 |
| Prakasam/Coastal | 1,231 (42.4) | 1,674 (57.6) | 2,905 |
| **Age Group** | | | |
| 40–49 years | 258 (6.5) | 3,738 (93.5) | 3,996 |
| 50–59 years | 607 (24.9) | 1,826 (75.1) | 2,433 |
| 60–69 years | 1,186 (60.0) | 791 (40.0) | 1,977 |
| 70+ years | 1,180 (90.1) | 130 (9.9) | 1,310 |
| **Gender** | | | |
| Male | 1,426 (32.3) | 2,990 (67.7) | 4,416 |
| Female | 1,805 (34.1) | 3,495 (65.9) | 5,300 |
| **Smoking** | | | |
| Yes | 1,855 (30.2) | 4,280 (69.8) | 6,135 |
| No | 1,372 (38.4) | 2,203 (61.6) | 3,575 |
| **Indoor kitchen smoke exposure:** | | | |
| Yes | 976 (34.9) | 1,817 (65.1) | 2,793 |
| No | 2,255 (32.6) | 4,668 (67.4) | 6,923 |
| **Lifetime cumulative effective sun exposure** | | | |
| 1st quintile | 324 (16.7) | 1,621 (83.3) | 1,945 |
| 2nd quintile | 370 (19.0) | 1,573 (81.0) | 1,943 |
| 3rd quintile | 530 (27.3) | 1,413 (72.7) | 1,943 |
| 4th quintile | 717 (36.9) | 1,225 (63.1) | 1,942 |
| 5th quintile | 1,287 (66.4) | 651 (33.6) | 1,938 |

and exposure to indoor smoke. (β:0.3; CI: 0.1–0.5). Cortical cataract showed a positive association with the 5th quintile of sun exposure (β: 2.6; CI: 1.0–4.2). Nuclear cataract showed a positive association with the sun exposure in 3rd (β:0.5, CI:0.2–0.7),4th (β: 0.9, CI: 0.7–1.1) and 5th quintile (β:2.1, CI:1.8–2.3) of sun exposure. Guwahati showed a higher chance of development of nuclear cataract (β: 1.6, CI: 1.4–1.9). Prakasam showed a high chance of development of nuclear (β: 1.9, CI: 1.7–21.1) and PSC (β: 0.8, CI:0.2–1.4). Out of the 9,710 people whose smoking status was known, 6135 were smokers and 1855 (30.2%) out of these had cataract. On multivariate analysis, smoking showed a positive association with cataract (OR: 1.4; CI: 1.2–1.6). Out of 9,716 people, 2793 (29%) were exposed to smoke in indoor kitchen. Out of these, 976 (34.9%) had cataract.

## Ultraviolet and aerosol exposure

**Daily integrated UV irradiance.** The long-term UV irradiance integrated in the range 280-400nm was studied using satellite data during the period from 1979–2005 over the entire Indian region which showed monthly or seasonal variability but does not show any significant change in the long-term. Generally, the climatology of UV shows a systematic latitudinal decrease from South to North, except when it reaches the high altitudes near the Himalayan region. In order to see the variation in UVA and UVB flux during the recent years, the CERES derived satellite data for daily mean were validated with the actual measurements at Delhi

**Table 3. Pattern of cataract among eyes evaluated using LOCS III.**

| | Delhi/Plain n (%) | Guwahati/Hills n (%) | Prakasam/Coastal n (%) | All Centres n (%) |
|---|---|---|---|---|
| | n = 1047 | n = 1151 | n = 1277 | n = 3475[*] |
| **Eyes with any** | | | | |
| PSC cataract | 454 (43.4) | 87 (7.6) | 40 (3.1) | 581 (16.7) |
| Nuclear cataract | 982 (93.8) | 1099 (95.4) | 1210 (94.7) | 3291 (94.7) |
| Cortical cataract | 546 (52.1) | 405 (35.2) | 30 (2.3) | 981 (28.2) |
| **Combinations of subtypes of cataract** | | | | |
| PSC only | 33 (3.2) | 29 (2.5) | 39 (3.1) | 101 (2.9) |
| Nuclear only | 335 (32.0) | 683 (59.3) | 1208 (94.6) | 2226 (64.1) |
| Cortical only | 21 (2.0) | 23 (2.0) | 27 (2.1) | 71 (2.0) |
| Nuclear + PSC | 133 (12.7) | 34 (3.0) | 0 (0.0) | 167 (4.8) |
| Cortical + PSC | 11 (1.1) | 0 (0.0) | 1 (0.1) | 12 (0.4) |
| Cortical + Nuclear | 237 (22.6) | 358 (31.1) | 2 (0.2) | 597 (17.2) |
| Cortical + Nuclear + PSC | 277 (26.5) | 24 (2.1) | 0 (0.0) | 301 (8.7) |

[*]In addition, 13624 eyes were normal. LOCS = Lens Opacities Classification System, PSC = Posterior subcapsular cataract

during October 2012 to September 2015. The daily mean values of UVA fluxe ranged ~ 1.1–20.1 Wm$^{-2}$ whereas UVB ranged ~ 0.03–0.53 Wm$^{-2}$. Seasonally, UVA and UVB radiation at Delhi, showed maximum during summer (~14 Wm$^{-2}$ for UVA & ~0.33 Wm$^{-2}$ for UVB) and

**Table 4. Multiple logistic regression showing association of cataract with various risk factors.**

| | Delhi/ Plain OR (95% CI) | Guwahati /Hills OR (95% CI) | Prakasam /Coastal OR (95% CI) | All Centres OR (95% CI) |
|---|---|---|---|---|
| | N = 3587 | N = 3207 | N = 2902 | N = 9696 |
| **Gender** | | | | |
| Male | 1 | 1 | 1 | 1 |
| Female | **0.6 (0.4, 0.8)**[+] | 1.2 (0.9, 1.5) | 1.0 (0.6, 1.5) | 0.9 (0.8, 1.1) |
| **Smoking** | | | | |
| No | 1 | 1 | 1 | 1 |
| Yes | **1.4 (1.1, 1.6)**[*] | **1.5 (1.1, 1.9)**[*] | **1.4 (1.1, 1.8)**[*] | **1.4 (1.2, 1.6)**[+] |
| **Indoor Smoke** | | | | |
| No | 1 | 1 | 1 | 1 |
| Yes | **1.3 (1.0, 1.8)**[*] | 1.3 (0.9, 1.8) | 1.4 (0.9, 2.1) | **1.2 (1.0, 1.4)**[*] |
| **Lifetime cumulative effective sun exposure** | | | | |
| 1st quintile | 1 | 1 | 1 | 1 |
| 2nd quintile | 1.2 (0.8, 1.8) | **2.4 (1.8, 3.2)**[+] | **0.5 (0.4, 0.8)**[*] | 1.1 (0.9, 1.3) |
| 3rd quintile | 1.3 (0.9, 1.9) | **6.5 (4.9, 8.7)**[+] | **0.5 (0.3, 0.6)**[+] | **1.6 (1.4, 1.9)**[+] |
| 4th quintile | **2.2 (1.6, 3.0)**[+] | **10.6 (7.6, 14.6)**[+] | 1.1 (0.8, 1.3) | **2.6 (2.2, 3.1)**[+] |
| 5th quintile | **7.5 (5.5, 10.4)**[+] | **25.7 (15.2, 43.5)**[+] | **5.0 (3.8, 6.5)**[+] | **9.4 (7.9, 11.2)**[+] |
| **Site** | | | | |
| Delhi/Plain | - | - | - | 1 |
| Guwahati/Hills | - | - | - | **2.0 (1.7, 2.3)**[+] |
| Prakasam/Coastal | - | - | - | **2.0 (1.8, 2.3)**[+] |

Only participants with any cataract assessed on clinical evaluation (including pseudophakia and aphakia) were included as cases and participants with no cataract were included as controls. Adjusted for educational status; OR = Odds Ratio; CI = Confidence Interval;

[*] p <0.05

[+] p<0.001

**Table 5. Multi-nomial logistic regression showing association of sub-types of cataract with various risk factors[*].**

| | Cortical β coefficient, (95% CI) | Nuclear β coefficient, (95% CI) | PSC β coefficient, (95% CI) |
|---|---|---|---|
| | (N = 7935) | (N = 7935) | (N = 7935) |
| **Gender** | | | |
| Male | Reference | Reference | Reference |
| Female | -0.5 (-1.6, 0.6) | -0.1 (-0.3, 0.1) | 0.1 (-0.8, 0.9) |
| **Smoking** | | | |
| No | Reference | Reference | Reference |
| Yes | 0.1 (-0.7, 0.9) | **0.4 (0.2, 0.6)** | 0.4 (-0.3, 1.0) |
| **Indoor Smoke** | | | |
| No | Reference | Reference | Reference |
| Yes | 0.1 (-1.0, 1.2) | **0.3 (0.1, 0.5)** | 0.4 (-0.4, 1.2) |
| **Lifetime Cumulative Effective Sun Exposure** | | | |
| 1st quintile | Reference | Reference | Reference |
| 2nd quintile | 1.3 (-0.2, 2.9) | 0.1 (-0.2, 0.3) | 0.2 (-0.5, 1.0) |
| 3rd quintile | 1.0 (-0.6, 2.7) | **0.5 (0.2, 0.7)** | 0.1 (-0.8, 0.9) |
| 4th quintile | 1.3 (-0.4, 2.9) | **0.9 (0.7, 1.1)** | 0.5 (-0.3, 1.3) |
| 5th quintile | **2.6 (1.0, 4.2)** | **2.1 (1.8, 2.3)** | 0.7 (-0.2, 1.6) |
| **Location** | | | |
| Delhi / Plain | Reference | Reference | Reference |
| Guwahati / Hilly | 0.4 (-0.6, 1.5) | **1.6 (1.4, 1.9)** | 0.1 (-0.7, 0.8) |
| Prakasam / Coastal | 0.6 (-0.2, 1.5) | **1.9 (1.7, 2.1)** | **0.8 (0.2, 1.4)** |

Adjusted for Education. Only participants with specific types of cataract were included as cases and participants with no cataract were included as controls.

CI = Confidence Interval;

[*] p <0.05

[+] p<0.001

minmum (~7 $Wm^{-2}$ & 0.12 $Wm^{-2}$) during winter. The annual mean values of UVA and UVB flux along with the AOD has been tabulated in **Table 6** for all the three stations, Delhi, Guwahati and Vishakhapatnam.

   **Air pollution exposure.**   The major air pollutants in the region are surface $SO_2$, $NO_2$, $PM_{10}$, $PM_{2.5}$ and surface ozone. Concentrations of the gaseous pollutants are generally within the National Ambient Air Quality Standards (NAAQS) but particulate matter ($PM_{10}$ and $PM_{2.5}$) is the major problem in all these cities which is significantly higher than the NAAQS values. $SO_2$ concentration is below NAAQS limits but $NO_2$ values are close to the limits at few places in these cities. At Guwahati the $SO_2$ and $NO_2$ concentration is far below NAAQS limits, SPM values are close to the limits but RSPM is exceeding the NAAQS limits. The long term observations indicate significant increasing trend in pollutants concentration in all the three locations.[10]

## Discussion (Table 7)

Cataract is a major cause of visual impairment in many low-income countries. This was a population based cross sectional study covering three diverse geographical regions in India in which the effect of various risk factors especially sun exposure on cataract has been studied in the people aged more than 40 years.

   The prevalence of cataract in age group more than 70 years was 90.1%, similar to the comparable age group in the INDEye study,[11] ACES,[12] Tanjong Pagar study[13] and the

**Table 6. Yearly averaged UVA, UVB flux in Wm$^{-2}$ and Aerosol Optical Depth (AOD) at three locations during 2010 to 2016.**

| Station Name | | Delhi/Plain | | | Guwahati/Hilly | | | Vishakhapatnam/Coastal | | |
|---|---|---|---|---|---|---|---|---|---|---|
| Year | Parameters | UVA | UVB | AOD | UVA | UVB | AOD | UVA | UVB | AOD |
| 2010 | Average | 10.66 ± 3.94 | 0.24 ± 0.11 | 0.66 ± 0.38 | 10.70 ± 3.08 | 0.26 ± 0.10 | 0.47 + 0.29 | 12.55 ± 3.60 | 0.33 ± 0.10 | 0.43 ± 0.19 |
| | Min. | 2.44 | 0.04 | 0.01 | 4.28 | 0.10 | 0.05 | 2.49 | 0.06 | 0.08 |
| | Max. | 18.00 | 0.50 | 2.57 | 18.77 | 0.52 | 1.91 | 18.95 | 0.55 | 1.52 |
| 2011 | Average | 10.66 ± 3.71 | 0.24± 0.11 | 0.66 ± 0.40 | 11.14 ± 3.26 | 0.27 ± 0.11 | 0.51 ± 0.29 | 12.95 ± 3.20 | 0.34± 0.09 | 0.47± 0.21 |
| | Min. | 2.92 | 0.05 | 0.14 | 2.57 | 0.05 | 0.08 | 2.11 | 0.04 | 0.14 |
| | Max. | 18.15 | 0.48 | 2.67 | 19.18 | 0.51 | 1.75 | 18.87 | 0.54 | 1.51 |
| 2012 | Average | 10.83 ± 3.78 | 0.25 ± 0.11 | 0.67 ± 0.38 | 10.84 ± 3.53 | 0.27 ± 0.11 | 0.54 ± 0.32 | 12.84 ± 3.01 | 0.34 ± 0.09 | 0.46 ± 0.22 |
| | Min. | 2.60 | 0.04 | 0.13 | 3.50 | 0.07 | 0.06 | 2.31 | 0.06 | 0.08 |
| | Max. | 18.01 | 0.47 | 2.27 | 18.80 | 0.53 | 1.72 | 19.80 | 0.56 | 1.52 |
| 2013 | Average | 10.92 ± 3.87 | 0.25 ± 0.11 | 0.64 ± 0.38 | 11.23± 3.33 | 0.28 ± 0.11 | 0.49 ± 0.36 | 13.05 ± 3.48 | 0.35 ± 0.10 | 0.46 + 0.19 |
| | Min. | 2.82 | 0.04 | 0.04 | 3.31 | 0.07 | 0.02 | 3.09 | 0.07 | 0.04 |
| | Max. | 18.19 | 0.49 | 2.39 | 20.54 | 0.54 | 2.27 | 18.83 | 0.53 | 1.46 |
| 2014 | Average | 10.89 ± 4.00 | 0.25 ± 0.12 | 0.66 ± 0.35 | 10.92 ± 3.25 | 0.26± 0.10 | 0.49 ± 0.30 | 12.55 ± 3.35 | 0.33± 0.10 | 0.52 + 0.22 |
| | Min. | 1.86 | 0.03 | 0.09 | 3.50 | 0.07 | 0.06 | 2.67 | 0.07 | 0.09 |
| | Max. | 18.32 | 0.46 | 2.50 | 18.68 | 0.53 | 1.49 | 19.80 | 0.54 | 1.25 |
| 2015 | Average | 10.75 ± 4.09 | 0.24 ± 0.12 | 0.65 ± 0.32 | 10.79 ± 3.36 | 0.25 ± 0.10 | 0.53 ± 0.34 | 12.92 ± 3.20 | 0.34 ± 0.09 | 0.47 + 0.22 |
| | Min. | 2.38 | 0.04 | 0.11 | 3.16 | 0.06 | 0.06 | 2.88 | 0.06 | 0.07 |
| | Max. | 18.85 | 0.47 | 2.01 | 19.72 | 0.54 | 1.98 | 19.50 | 0.54 | 1.46 |
| 2016 | Average | 11.29 ± 3.93 | 0.26 ± 0.12 | 0.71 ± 0.39 | 10.27 ± 3.77 | 0.25 ± 0.12 | 0.51 ± 0.36 | 13.53 ± 3.33 | 0.38 ± 0.11 | 0.49 + 0.20 |
| | Min. | 2.82 | 0.04 | 0.08 | 2.59 | 0.04 | 0.05 | 2.57 | 0.07 | 0.14 |
| | Max. | 17.98 | 0.46 | 3.51 | 18.67 | 0.54 | 1.75 | 19.64 | 0.58 | 1.50 |

UVA = Ultraviolet radiation A, UVB = Ultraviolet radiation B

Metilka eye study.[14] It is also observed in the present study that prevalence of cataract in males and females was similar, while previous studies have reported higher prevalence in females.[11] The site closest to the equator (Prakasam, 15.4 degrees N; 42.4%) had the highest rate of cataract reported than the other two sites which were located at similar latitudes (Guwahati, 26.1 degrees N; 26.6%, Delhi, 28.7 degrees N; 31.9%) and between the latter two, the rate was higher at Delhi than Guwahati. Similar findings were reported by Javitt JC et al who found that latitude correlates directly with the amount of UV-B radiation and the probability of cataract surgery in the U.S. increases 3% for every 1 degree decrease in latitude.[15] The present study highlights that while latitude is indeed an important factor, there are other factors that influence development of cataract as well.

What is interesting is that the median lifetime sun exposure was higher in Delhi (114140) than in Prakasam (109889), and in both of these sites it was nearly one and a half times that in Guwahati (72 759). This suggest that disparities in the nature of sun exposure such as work on water, variations in UV exposure at different elevations, and further differences due to living and working in proximity to an urban centre versus a more rural location associated with each geographical site have additional influences and roles to play in development of cataract.

Through physical measurements of the UV-A and UV-B, it was observed that ambient radiation levels have remained constant over these regions since a last four decades and are higher in coastal areas (Table 6). The higher prevalence of cataract in coastal area as compared to plains and hilly populations may be partly explained by these differences in UV radiation levels. Higher odds of cataract were observed in hills compared to plains, while both areas have similar UV levels. Another potential factor could be the relationship between solar angle and

**Table 7. Comparison of our study with previously reported studies.**

| Authors/Journal/Year of Publication | Type of study | Sun exposure | Smoking | Indoor kitchen smoke exposure | Gender | Education and Occupation | Remarks |
|---|---|---|---|---|---|---|---|
| Present study | Population based cross sectional | Increased risk of any cataract and nuclear and cortical cataract | Increased risk of any cataract and nuclear cataract | Increased risk of any cataract and nuclear cataract | M = F | Not assessed | Nil |
| Beltran-Zambrano E et al. Arch Soc Espanola Oftalmol 2019 [18] | Meta-analysis | Not assessed | Increased risk of any cataract and nuclear cataract | Not assessed | Not assessed | Not assessed | Nil |
| Ravilla TD et al. Environ Health Perspect 2016 [4] | Population based study North and south India | Increased risk of any cataract | Increased risk of any cataract | Increased risk of cataract in women, nuclear and posterior sub capsular cataract | Not assessed | Not assessed | Nil |
| Tang et al. PLoS One 2015 [2] | Population based cross sectional study | UV-B = +: Cortical cataract | Not assessed | Not assessed | F > M Increased risk of nuclear and cortical cataract | Outdoor activity increased risk of cortical cataract | High myopia increased risk of nuclear cataract |
| Lindblad et al JAMA 2014 [19] | Prospective Cohort | Not assessed | Increased risk of risk of cataract extraction | Not assessed | Not assessed | Not assessed | Smoking cessation decreased risk of cataract extraction |
| Ye J et al. Invest Ophthalmol Vis Sci 2012 [20] | Meta-analysis | Not assessed | Increased risk of any cataract, nuclear and posterior sub capsular cataract | Not assessed | Not assessed | Not assessed | Nil |
| Vashisht P et al. Ophthalmology 2011 [21] | Population based cross sectional study | Not assessed | Not assessed | Not assessed | F > M | Not assessed | Nuclear cataract most prevalent cataract |
| Athanasiov PA et al. Br J Ophthalmol 2008 [14] | Population based cross sectional study | Not assessed | No association | Not assessed | No association | Low education increased risk of nuclear cataract Outdoor occupation: No association | Betel nut use: No association with cataract |
| Pastor-Valero M et al. BMC Ophthalmol 2007 [22] | Case control study | No association between sun exposure over adult life and cataract | Not assessed | Not assessed | Not assessed | Outdoor exposure: No association Outdoor exposure at young age increased risk of nuclear cataract | Nil |
| Raju P et al. Br J Ophthalmol 2006 [23] | Population based study | Not assessed | Smoking: no association Smokeless tobacco +: NC | Not assessed | Not assessed | Not assessed | Nil |
| Pokhrel AK et al. Int. J Epidemiol 2005 [24] | Hospital based case control | Not assessed | Not assessed | Solid fuel in unflued stoves increased risk of cataract in women | Not assessed | Not assessed | Lack of kitchen ventilation: risk factor for cataract |
| Krishnaiah S et al. Invest Ophthalmol Vis Sci 2005 [25] | Population based cross sectional study | Not assessed | Increased risk of cataract, nuclear and cortical cataract | Not assessed | Higher prevalence in women | Increased risk with illiterates and lower socio-economic group | Nil |
| Nirmalan PK et al. Br J Ophthalmol 2004 [26] | Population based cross sectional study | Not assessed | Increased risk of nuclear and cortical cataract | Not assessed | F > M Increased risk of nuclear and cortical cataract | Illiteracy is a risk factor | Nil |

(*Continued*)

**Table 7.**  (Continued)

| Authors/Journal/ Year of Publication | Type of study | Sun exposure | Smoking | Indoor kitchen smoke exposure | Gender | Education and Occupation | Remarks |
|---|---|---|---|---|---|---|---|
| Nirmalan PK et al. Invest Ophthalmol Vis Sci 2003 [27] | Population based cross sectional study | Not assessed | Not assessed | Not assessed | F > M | Not assessed | Most common subtype was nuclear cataract |
| Neale et al. Epidemiol Camb Mass 2003 [5] | Case control | +: NC | +: NC | Not assessed | Women more likely to have nuclear cataract | Education beyond school decreased risk | Diabetes increased risk of cataract |
| Seah et al. Ophthalmology 2002 [13] | Population based cross sectional study | Not assessed | Not assessed | Not assessed | M = F | Not assessed | Most common subtype was nuclear cataract |
| Delcourt e al. Arch Ophthalmol. 2000 [28] | Population based study | Increased risk of cataract and cortical cataract | Not assessed | Not assessed | Not assessed | Not assessed | Nil |
| West SK et al. JAMA 1998[29] | Population based cohort | UV-B Increased risk of Cortical cataract | Not assessed | Not assessed | Not assessed | Not assessed | Nil |
| Wong L et al. J Epidemiol Community Health 1993 [30] | Cross sectional survey | Increased risk of cataract and nuclear cataract | Not assessed | Not assessed | M = F | Not assessed | Nil |
| Taylor HR et al. NEJM 1988 [31] | Epidemiological survey | UV-B Increased risk of Cortical cataract | Not assessed | Not assessed | Not assessed | Not assessed | Nil |

sun exposure. As demonstrated by Sasaki et al, ocular exposure to UVR is effectually maximum at times when sun protection is less likely to be used, such as in the mid-morning, mid-afternoon; and in the fall, winter, and spring.[16] These findings merit subsequent exploration via longitudinal studies and measurements of individual levels of UV exposure to the eyes.

An important finding in the present study is that sun exposure is a risk factor for cataract especially cortical and nuclear cataract but not PSC. A dose response relationship was observed wherein increasing levels of sun exposure were associated with higher odds of cataract. Similar findings were seen in Taizhou eye study done in rural Chinese population aged $\geq$ 45 years.[2] Collman et al reported the significant association of cortical cataract and PSC with sun exposure.[3] Similar positive association of cortical cataract were reported by Hiller et al using the 1971–1972 National Health and Nutrition Examination Survey dataset and Delcourt from France. [17]

Wong et al evaluated the association between sun exposure, antioxidant status and cataract in Hong Kong fisherman and reported higher grades of cataract in people having higher sun exposure particularly nuclear cataract was reported.[30] The present study also found a strong association between increasing sun exposure and cataract especially nuclear and cortical cataract. PSC does not show any association with sun exposure. Taylor et al reported increasing risk of cortical cataract with high cumulative levels of UVB radiation while no association was found between nuclear cataracts and ultraviolet B exposure or between cataracts and ultraviolet A exposure.[31] Similar results were seen in the Beaver Dam Eye Study wherein men with higher levels of UVB light exposure were 1.36 times more likely to have more severe cortical opacities.[32] However, UVB exposure was not found to be associated with nuclear sclerosis or posterior subcapsular opacities. Also, women did not show any association with UVB exposure as they were less likely to be exposed to UVB. West et al determined the ocular exposure to UV-B radiation for a population of older persons and found the odds of cortical cataract to increase with increasing ocular exposure to UV-B.[29] Neale et al further found a strong

positive association of occupational sun exposure between the ages of 20 and 29 years with nuclear cataract rather than for exposure later in life.[5] Similarly, María et al reported no association between outdoor exposure and risk of cataract.[22] However, they also observed a positive association between years of outdoor exposure at younger ages and risk of nuclear cataract later in life.

This study shows a positive association between smoking and cataract especially nuclear cataract. It corroborates with results of previous studies which correlate smoking with cataract. Krishnaiah et al found a significantly higher prevalence of nuclear, cortical cataract, and history of prior cataract surgery and/ or total cataract among cigarette smokers.[25] Raju et al in their study report some interesting results as in they did not find any association of smoking with cataract but the use of smokeless tobacco use was found to be more strongly associated with cataract especially nuclear cataract.[23] Similarly, Ye et al reported current smokers to be at a higher risk of cataract especially nuclear cataract as compared to past smokers in their meta-analysis.[33] An association between smoking and cataract, especially nuclear cataract, was observed. No association was seen between smoking and cortical or PSC. Lindblad et al further reported that smoking cessation decreases the risk of cataract extraction with time, although the risk persists for decades.[19] The higher the intensity of smoking, the longer it takes for the increased risk to decline. These findings emphasize the importance of early smoking cessation and preferably the avoidance of smoking. The association between smoking and nuclear cataract has also been shown by Beltrán-Zambrano et al and Christen et al. [34]

Biomass cooking fuels are very common in Indian households, especially by the lower socioeconomic and rural population, but there is a limited evidence on association between cataract and biomass fuels. [4] Thulasiraj et al examined the association of biomass cooking fuels with cataract in a population-based study done in north and south India in people $\geq 60$ years old and provides evidence for the association of biomass fuels with cataract for women but not for men. [4] Inhalation of the pollutants released by indoor kitchen fuels may cause deleterious effects on health. Sukhsohale et al reported symptoms like eye irritation, headache, and diminution of vision to be significantly higher in biomass users.[35] Pokhrel et al found that the prevalence of cataract is higher especially in females and provides confirmatory evidence that use of solid fuel in indoor stoves is associated with increased risk of cataract in women who do the cooking. [24] In the present study, out of 9,716 people evaluated for presence of indoor kitchen, 2793 had smoke exposure, out of which 976 (34.9%) developed cataract. Using indoor kitchen was found to be significantly associated with the occurrence of cataract especially nuclear cataract.

The main strength of the present study is extensive nature of the work done, over a large population, across three different geographical locations in India, evaluating the association between cataract and various risk factors. Mixed cataracts were excluded when estimating the association of subtypes of cataract with the risk factors. Standardized methods were used to measure outdoor exposure, and to assess the cataract using LOCS III grading. Time spent outdoors was used as a proxy for UV-A and UV-B exposure which is similar to the approach taken by prior studies. Physical measurements of UV exposure and environmental parameters were done at all the sites, parameters which have not been reported by prior studies on sun exposure and cataract from developing countries.

This study has a few limitations. A large proportion of the patients were pseudophakic or aphakic thereby precluding LOCS III assessments in all eyes defined for cataract. This being a cross-sectional study, the data is prone to recall bias especially with regards to history of risk factors. The nature of headgear used may have an influence on the amount of UV protection accorded by it and this too needs exploration in the context of locally relevant headgear. The study did not capture detailed information on the designs of headgear, or the particular eye

wear such as type of sunglasses, spectacles etc. None of the patients reported use of contact lenses, which was not surprising given that the study was done in rural Indian populations. It has been reported in literature that different types of headgear and sunglasses may accord widely varying levels of protection depending on the geometry, wearing position, head positions and exposure conditions. [36,37] There are seasonal variations in UV exposure at each site which we could not measure at an individual level. Another contributory factor would be living and working in proximity to an urban center vs. a more rural location. Work on water, which is reported to have a large impact on effective sun exposure and has been reported as an important risk factor for cataract [6,30,31], could be an additional contributory factor in Prakasam, which is a coastal district. This aspect needs further exploration in future studies.

Based on the results, it is recommended future studies should try to get in-depth information on nature of headgear used and the time of day when used, duration of work on water, and ascertain individual level UV exposures from participants recruited from diverse altitudes and latitudes. Finally, models of levels of UV protection accorded by different types of headgear and eye-wear used in India should be developed.

The most important association of cataract has been found with sun exposure. The study calculated the lifetime sun exposure of a person using Melbourne study formula which takes into account the number of hours spent in sun and the use of protective headgear. The median duration of sun exposure in plain, hilly and coastal region was 114140, 72759 and 109889 hours respectively. A higher percentage of cataract was seen in coastal areas (42.4%) followed by plain (31.9%) and hilly areas (26.6%). Sun exposure has shown a significant association with the development of cataract especially as the severity increases in 4th and 5th quintile. Cataract is also seen to be associated with smoking and exposure to smoke while working in indoor kitchen. Sun exposure shows a significant association with nuclear cataract and cortical cataract.

## Conclusions

The study is the first to look at the interplay of known risk factors with different geographic locations in India. It demonstrates clear influence of several modifiable risk factors such as sun exposure, smoking and indoor kitchen smoke exposure exploring environmental differences as well as establishing the importance of these behavioral risk factors in increasing the likelihood of cataract. Interventions that address these behaviors may be effective in decreasing logterm risk of cataract and merit further explorations in long term community trials. The information of likelihood varying with location is pertinent as it can further influence strategies for prevention and control.

## Supporting information

**S1 File. Questionnaires used in the study.** The Hindi and English versions of the questionnaires used in the study.
(PDF)

**S2 File. Data package.** The study dataset in Stata and spreadsheet formats, codebooks, analysis commands, output logs and instructions for using the dataset.
(ZIP)

## Acknowledgments

The authors would like to acknowledge the contributions of—The Indian Council of Medical Research–Eye Sun Exposure & Environment "ICMR–EYE SEE" Study Group, with lead

author—Dr. Radhika Tandon (radhika_tan@yahoo.com) Dr. Rajendra Prasad Centre for Ophthalmic Sciences, AIIMS, New Delhi. The members of the group include (a) Dr. Rajendra Prasad Centre for Ophthalmic Sciences, AIIMS, New Delhi: Dr. Radhika Tandon, Dr. Praveen Vashist, Dr. Noopur Gupta, Dr. Vivek Gupta, Dr. Rashmi Singh, Dr. Meenakshi Wadhwani, Dr. Shweta, Dr. Aparna Gupta, Dr. Saurabh Agarwal Jwalaprasad, Dr. Bhagbat Nayak; (b) Public Health Foundation of India, Hyderabad: Dr. GVS Murthy; (c) Pushpagiri Vitreo Retina Institute, Secunderabad: Dr. K. Vishwanath; (d) Regional Institute of Ophthalmology, Guwahati: Dr. C.K. Barua, Dr. Dipali Deka, Dr. Jayanta Thakuria, Dr. Indrani Goswami; and (e) National Physical Laboratory, New Delhi: Dr Sachchidanand Singh, Ms. Tanya Patel, Ms. Ankita Mall, Dr. Rupesh M Das. Mr Amit Bhardwaj and Mr Deepak Kumar are acknowledged for their contribution to data management and analysis. We also acknowledge the ICMR Task Force on Global Climate Change and Health chaired by Prof. Seyed E. Hasnain, IIT Delhi, for periodic review and technical inputs during the course of the study.

## Author Contributions

**Conceptualization:** Praveen Vashist, Radhika Tandon, G. V. S. Murthy, Sachchidanand Singh.

**Formal analysis:** Praveen Vashist, Radhika Tandon, G. V. S. Murthy, Sachchidanand Singh, Vivek Gupta, Noopur Gupta, Meenakshi Wadhwani, Rashmi Singh.

**Funding acquisition:** Radhika Tandon.

**Investigation:** Praveen Vashist, Radhika Tandon, G. V. S. Murthy, C. K. Barua, Dipali Deka, Sachchidanand Singh, Vivek Gupta, Noopur Gupta, Meenakshi Wadhwani, Rashmi Singh, K. Vishwanath.

**Methodology:** Praveen Vashist, Radhika Tandon, G. V. S. Murthy, C. K. Barua, Dipali Deka, Sachchidanand Singh, K. Vishwanath.

**Supervision:** Praveen Vashist, Radhika Tandon, G. V. S. Murthy, C. K. Barua, Dipali Deka, Sachchidanand Singh, K. Vishwanath.

**Writing – original draft:** Praveen Vashist, Radhika Tandon, G. V. S. Murthy, Sachchidanand Singh, Vivek Gupta, Noopur Gupta, Meenakshi Wadhwani, Rashmi Singh.

**Writing – review & editing:** Praveen Vashist, Radhika Tandon, G. V. S. Murthy, C. K. Barua, Dipali Deka, Sachchidanand Singh, Vivek Gupta, Noopur Gupta, Meenakshi Wadhwani, Rashmi Singh, K. Vishwanath.

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
