## [Decision Letter · Decision Letter 0]

1 Nov 2019

PONE-D-19-22134

“Association of Cataract and Sun Exposure in geographically diverse populations of India: The CASE study. First Report of the ICMR-EYE SEE (Indian Council of Medical Research-Eye Sun Exposure and Environment) Study Group”

PLOS ONE

Dear Dr. Tandon,

Thank you for submitting your manuscript to PLOS ONE. After careful consideration, we feel that it has merit but does not fully meet PLOS ONE’s publication criteria as it currently stands. Therefore, we invite you to submit a revised version of the manuscript that addresses the points raised during the review process.

We would appreciate receiving your revised manuscript by Dec 16 2019 11:59PM. To enhance the reproducibility of your results, we recommend that if applicable you deposit your laboratory protocols in protocols.io, where a protocol can be assigned its own identifier (DOI) such that it can be cited independently in the future. For instructions see: http://journals.plos.org/plosone/s/submission-guidelines#loc-laboratory-protocols

We look forward to receiving your revised manuscript.

Kind regards,

James Wolffsohn, PhD

Academic Editor

PLOS ONE

Journal Requirements:

2. Please include additional information regarding the survey or questionnaire used in the study and ensure that you have provided sufficient details that others could replicate the analyses. For instance, if you developed a questionnaire as part of this study and it is not under a copyright license more restrictive than CC-BY, please include a copy, in both the original language and English, as Supporting Information.

3, We note that you have indicated that data from this study are available upon request. PLOS only allows data to be available upon request if there are legal or ethical restrictions on sharing data publicly. For information on unacceptable data access restrictions, please see http://journals.plos.org/plosone/s/data-availability#loc-unacceptable-data-access-restrictions.

5. Please amend the manuscript submission data (via Edit Submission) to include authors GVS Murthy , CK Barua , Dipali Deka , Sachchidanand Singh, Vivek Gupta , Noopur Gupta , Meenakshi Wadhwani , Rashmi Singh and K Vishwanath

6. One of the noted authors is a group or consortium: ICMR-EYE SEE Study Group. In addition to naming the author group, please list the individual authors and affiliations within this group in the acknowledgments section of your manuscript. Please also indicate clearly a lead author for this group along with a contact email address.

Additional Editor Comments:

Well done on putting together such a good manuscript. Please could you address the minor comments and resubmit

Reviewers' comments:

Reviewer's Responses to Questions

**Comments to the Author**

1. Is the manuscript technically sound, and do the data support the conclusions?

Reviewer #1: Yes

Reviewer #2: Yes

2. Has the statistical analysis been performed appropriately and rigorously? 

Reviewer #1: Yes

Reviewer #2: I Don't Know

3. Have the authors made all data underlying the findings in their manuscript fully available?

Reviewer #1: Yes

Reviewer #2: Yes

4. Is the manuscript presented in an intelligible fashion and written in standard English?

Reviewer #1: Yes

Reviewer #2: Yes

5. Review Comments to the Author

Reviewer #1: Summary & Overall Impression

This was a very large, comprehensive and well conducted study of the relationship of sun exposure in India. The authors were able to amass a large sample of over 12,000 subjects spread relatively evenly across 3 distinct geographic areas of India, and to collect clinical and subjective data on them. They present their data in a clear and logical fashion, and have employed adequate statistical rigor to their work. Their conclusion that cataracts were associated with increasing levels of sun exposure, smoking and in some cases, exposure to indoor kitchen smoke is consistent with other published studies, and they have pointed out nuances relevant to the population and conditions in India. I find this to be a relevant and important addition to the literature, and recommend that it be published after addressing some minor issues with consistency in the paper.

Issues to be addressed (minor)

At line 311, they discuss noting increasing prevalence in cataract with decreasing distance to the equator. However, their data in Table 1 shows that while the site closest to the equator (Prakasam at ~15 degrees N) had the highest rate of cataract reported (42.4%), the site at ~28 degrees N (Guwahati - 31.9%) actually had a higher rate than the site at ~26 degrees N (Gurgaon - 26.6%). Additionally, when looking at the total UV exposures reported in Table 2, neither is the median lifetime UV exposure data consistent with the rates of cataracts. The authors should discuss additional potential factors such as work on water, increase in UV exposure at higher elevations and differences due to living and working in proximity to an urban center vs. a more rural location.

At line 328 , the authors cite 2 studies discussing the association with sun exposure and cataract, but fail to mention a very important environmental factor - working over water, as was the case with Hong Kong fisherman and the waterman of Chesapeake Bay. In the paper, which the authors have cited as a basis of their UV exposure calculations, McCarty shows a LARGE influence of water (a factor of 1.9 applied to over water exposure times). This may be an interesting factors for the authors to characterize in future studies, particularly in the regions with ocean and large river influences. This comes up again at line 347.

Another issue which should be considered is the influence of solar angle, as described by Sasaki, which can influence ocular exposure differently based on time of year and time of day. This was not considered in McCarty’s calculations but adds an interesting point of discussion… the latitudes nearer the equator always have higher ambient exposure, but more northern sites (or southern for the southern hemisphere) have direct ocular exposures that may be as high in early and late hours of summer days, and all day during spring and autumn (Sasaki H, Sakamoto, Schnider C et al. (2011). UV-B Exposure to the Eye Depending on Solar Altitude. Eye & contact lens. 37. 191-5.)

At line 403-405, the numbers quote for exposure are not consistent with those quoted in table 1 (means versus median, perhaps). And again, the reported association of higher cataract in coastal areas followed by plain and hilly areas is out of order compared to data in table 2.

Reviewer #2: the paper is interesting and helps add to the literature.

However, there are quite a few instances where the first person is used in the manuscript so I believe this should be addressed to amend to the third person.

There also needs to be a little more about UV protection worn (hats, sunglasses, CLs) - it is mentioned briefly but not explored further.

6. PLOS authors have the option to publish the peer review history of their article (what does this mean?). If published, this will include your full peer review and any attached files.

Reviewer #1: Yes: Cristina Schnider, OD, MSc, MBA, FAAO

Reviewer #2: No

---

## [Author Response · Author response to Decision Letter 0]

16 Dec 2019

Dear Dr Wolffsohn

We thank you and the reviewers for a comprehensive review of our manuscript referenced above and for providing positive feedback on how we can improve the paper. We have tried to address these comments in a revised version. 

We believe that the comments have greatly helped improve the quality of the manuscript and sincerely hope it now meets your criteria for publication. We have made few other corrections as well addressing clarity and formatting of the manuscript as per PLOS ONE guidelines.

The revised manuscript (marked-up and unmarked clean versions) as well as the point by point reply are enclosed herewith.

With Regards

Prof Radhika Tandon

---

## [Editor Report · Decision Letter 1]

2 Jan 2020

Association of Cataract and Sun Exposure in geographically diverse populations of India: The CASE study. First Report of the ICMR-EYE SEE Study Group

PONE-D-19-22134R1

Dear Dr. Tandon,

We are pleased to inform you that your manuscript has been judged scientifically suitable for publication and will be formally accepted for publication once it complies with all outstanding technical requirements.

With kind regards,

James Wolffsohn, PhD

Academic Editor

PLOS ONE

Additional Editor Comments (optional):

Thank you for comprehensively addressing the reviewers comments

---

## [Editor Report · Acceptance letter]

6 Jan 2020

PONE-D-19-22134R1 

Association of Cataract and Sun Exposure in geographically diverse populations of India: The CASE study. First Report of the ICMR-EYE SEE Study Group 

Dear Dr. Tandon:

I am pleased to inform you that your manuscript has been deemed suitable for publication in PLOS ONE. Congratulations! Your manuscript is now with our production department. 

With kind regards,

on behalf of

Professor James Wolffsohn 

Academic Editor

PLOS ONE